# High-density DArT-based SilicoDArT and SNP markers for genetic diversity and population structure studies in cassava (*Manihot esculenta* Crantz)

Bright Gyamfi Adu[1]*, Richard Akromah[2], Stephen Amoah[2], Daniel Nyadanu[3‡], Alex Yeboah[4‡], Lawrence Missah Aboagye[1‡], Richard Adu Amoah[1‡], Eva Gyamfuaa Owusu[5‡]

1 Council for Scientific and Industrial Research-Plant Genetics Resources Research Institute, Bunso, Ghana, 2 Department of Crop and Soil Sciences, Kwame Nkrumah University of Science and Technology, Kumasi, Ghana, 3 Cocoa Research Institute of Ghana, New Tafo-Akim, Ghana, 4 Council for Scientific and Industrial Research -Savanna Agricultural Research Institute, Tamale, Ghana, 5 Department of Statistics and Actuarial Sciences, Kwame Nkrumah University of Science and Technology, Kumasi, Ghana

☯ These authors contributed equally to this work.
‡ DN, AY, LMA, RAA and EGO also contributed equally to this work.
* adubright1992@gmail.com

**Data Availability Statement:** All relevant data are within the manuscript and its Supporting Information files.

## Abstract

Cassava (*Manihot esculenta* Crantz) is an important industrial and staple crop due to its high starch content, low input requirement, and resilience which makes it an ideal crop for sustainable agricultural systems and marginal lands in the tropics. However, the lack of genomic information on local genetic resources has impeded efficient conservation and improvement of the crop and the exploration of its full agronomic and breeding potential. This work was carried out to obtain information on population structure and extent of genetic variability among some local landraces conserved at the Plant Genetic Resources Research Institute, Ghana and exotic cassava accessions with Diversity Array Technology based SilicoDArT and SNP markers to infer how the relatedness in the genetic materials can be used to enhance germplasm curation and future breeding efforts. A total of 10521 SilicoDArT and 10808 SNP markers were used with varying polymorphic information content (PIC) values. The average PIC was 0.36 and 0.28 for the SilicoDArT and SNPs respectively. Population structure and average linkage hierarchical clustering based on SNPs revealed two distinct subpopulations and a large number of admixtures. Both DArT platforms identified 22 landraces as potential duplicates based on Gower's genetic dissimilarity. The expected heterozygosity which defines the genetic variation within each subpopulation was 0.008 for subpop1 which were mainly landraces and 0.391 for subpop2 indicating the homogeneous and admixture nature of the two subpopulations. Further analysis upon removal of the duplicates increased the expected heterozygosity of subpop1 from 0.008 to 0.357. A mantel test indicated strong interdependence (r = 0.970; P < 0.001) between SilicoDArT and DArTSeq SNP genotypic data suggesting both marker platforms as a robust system for genomic studies in cassava. These findings provide important information for efficient *ex-situ*

**Funding:** The author(s) received no specific funding for this work.

**Competing interests:** The authors have declared that no competing interests exist.

conservation of cassava, future heterosis breeding, and marker-assisted selection (MAS) to enhance cassava improvement.

## Introduction

Cassava (*Manihot esculenta* Crantz) is the third most important source of calories in the tropics after rice and maize with millions of people depending on it in the world [1, 2]. Mostly grown in marginal ecologies, the crop is usually cultivated by smallholder farmers due to its ability to grow and yield in unfavourable conditions with poor soil fertility and low rainfall [1, 3]. The starchy roots contain mainly carbohydrates and the leaves are also used as a vegetable in some African countries which are cheap but a rich source of proteins, vitamins A, B and C, and other minerals [3, 4]. The cultivation of the crop continues to spread in Ghana and around the globe due to its storage roots which serve as the main raw material for industrial starch and alcohol production [5, 6]. However, marginal yields continue to be realised at the farm level which is partly due to the lack of improved varieties and the use of low yielding and environmentally sensitive lines by farmers [7]. There are also new emerging and diversified markets demand for cassava in Ghana which further suggests breeding for improved cultivars to meet specific domestic and industrial needs [8].

Germplasm banks and farmer's fields serve as reservoirs of genetic variability of crops. These collections harbour genes or germplasm with the potential to improve productivity and adaptation or tolerance to abiotic and biotic stresses [9, 10] which is particularly relevant in the current frame of climatic change and global warming. It is therefore imperative to identify true biodiversity in biological resources for effcient management, including conservation and selection of genetically divergent accessions to optimize breeding programs [11]. Genetic diversity and population structure analysis are important for characterizing the natural selection history and genetic relationships among accessions [12]. A comprehensive understanding of the genetic variability and relationships in available germplasm is a determining factor towards efficient conservation and designing breeding programmes and/or achieving breeding objectives. Several reports have highlighted the significant genetic variability within the cassava genepool for several traits associated with yield, disease resistance and drought that can be exploited for crop improvement [3, 8, 13–16].

Diversity analysis is an important component of plant breeding and genetics, conservation and evolution [17]. Most cassava diversity analyses have been based on phenotypic characters [18–22] which are mostly not reliable and environmentally plastic [23–25]. Therefore, genomic techniques may be useful in diversity assessment and selection. The use of molecular tools in plant genetic analyses and crop improvement cannot be overemphasized [26–29]. Molecular tools have proven to be more reliable in identifying duplicates among accessions during characterization which is important towards revealing genuine variability for breeding and reducing space and maintenance cost at gene banks [30, 31]. Molecular marker technologies such as Random Amplified Polymorphic DNAs (RAPDs), allozymes, single nucleotide polymorphisms (SNPs), Simple Sequence Repeats (SSRs) are available for genetic study in cassava from Ghana and other countries [32–35]. However, several degrees of limitations are also associated with these gel-based molecular marker systems which include lack of reliability and resolution (RAPD markers), poor genome coverage or labour intensive and not amenable for throughput genotyping (AFLP, RFLP) or less cost-effectiveness or require sequence information (SSRs) [36, 37]. These factors limit their applicability for many crops, especially for 'orphan' crops and polyploid species [37]. To this end, DArT markers developed through Next

Generation Sequencing platforms (NGS) are the prime alternative for molecular studies since they cover wide section of the genome with high-throughput and cost effective [38].

DArT markers (Diversity Array Technology Pty Ltd) was developed as one of the ultra-high-throughput, sequence independent, cost effective, whole-genome genotyping technique with large number of markers that cover the entire genome [36]. DArT markers have been applied successfully in genomic studies in many species including those with large and complex genomes such as barley, sugarcane, wheat, oat and strawberry [37, 39–43]. DArT markers are developed through the use of combinations of restriction enzyme digestions to reduce genome complexity, followed by next-generation sequencing of complexity reduced representations or fragments to identify DNA polymorphisms and SNPs leading to the production of thousands of polymorphic loci in a single assay [38, 44, 45]. Currently, DArT platform generates two variants of markers (SilicoDArT and DArTSeq SNP markers). SilicoDArT markers are dominant and are mostly scored for the absence (0) or presence (1) of a single allele while as DArTSeq SNPs are co-dominant markers [38, 44].

Xia et al. [46] used DArT for high-throughput genotyping of cassava and its wild relatives and suggested *PstI/TaqI* and *PstI/ BstNI* as the best complexity reduction method. DArTSeq was recently used to generate a garlic core collection from the accessions kept in garlic germplasm bank, Cordoba, Spain by revealing that 31.5% of the accessions were genetically redundant [30]. These cassava germplasm have been phenotypically characterized, nonetheless, redundancy may be expected [26, 27]. Herein, the DArT markers (SilicoDArT and DArTseq SNPs) were used to analyse the genetic dissimilarities among a collection of cassava germplasm Ghana and examine the structure of the population. This will lay a foundation for efficient curation of cassava germplasm and parental selection in future breeding programs.

## Materials and methods

### Plant materials

A total of 87 cassava accessions were used in the study (**Table 1**). Sixty-two (62) of these were local cultivars comprising of local landraces (41) and locally improved/released (21) cultivars. The landraces were obtained from CSIR-Plant Genetic Resource and Research Institute of Ghana (PGRRI-Bunso). The improved cultivars were also obtained from CSIR-Crops Research Institute (Kumasi, Ghana). The remaining 25 exotic genotypes including drought tolerant populations from International Institute of Tropical Agriculture, Nigeria (IITA) were obtained through CSIR-Savanna Agricultural Research Institute, Nyankpala (CSIR-SARI, Ghana). The 87 cassava genotypes were established on the field for morphological characterization using a standard cassava descriptor in the year 2018 [25].

### Molecular characterization

Extraction and quantification of DNA were carried out at the Agricultural Biotechnology Laboratory, Faculty of Agriculture, Kwame Nkrumah University of Science and Technology, Ghana while the genotyping by sequencing was done at the Diversity Array Technology Laboratory, University of Canberra, Australia.

**Genomic DNA extraction.** DNA samples were extracted from the youngest fully expanded leaves of each of the 87 cassava genotypes two weeks after planting using the DArT DNA extraction protocol [38]. The concentration of extracted DNA was checked using the Nanodrop 2000c spectrophotometer (NanoDrop Lite, LT2878, Thermo Scientific, USA). DNA samples were diluted between 50–100 ng/µl based on the Nanodrop readings. The samples were then packaged and shipped to Diversity Array Technology incorporation, Australia for DArTSeq genotyping.

**Table 1. List of cassava accessions used in the study and some morphological attributes.**

| Accession Number | Name | Source/region of collection | Colour of apical leaves | Average % roots dry matter |
|---|---|---|---|---|
| 1 | Sika bankye | Locally released variety | Purplish green | 34.67 |
| 2 | UCC-01-464 | Local, Landrace, Central Ghana | Purplish green | 35.78 |
| 3 | DMA-00-024 | Local, Landrace, Brong Ahafo, Ghana | Dark green | 41.00 |
| 4 | UCC-01-144 | Local, Landrace, Central Ghana | Purplish green | 40.68 |
| 5 | 1090090 | IITA | Purplish green | 37.45 |
| 6 | DMA-00-070 | Local, Landrace, Brong Ahafo, Ghana | Dark green | 41.03 |
| 7 | IBA010040 | IITA | Light green | 38.55 |
| 8 | IBA070134 | IITA | Dark green | 40.99 |
| 9 | UCC-01-507 | Local, Landrace, Central Ghana | Purple | 40.57 |
| 10 | UCC-01-015 | Local, Landrace, Central Ghana | Dark green | 33.21 |
| 11 | BD-96-065 | Local, Landrace, Eastern, Ghana | Dark green | 39.92 |
| 12 | KSI-00-191 | Local, Landrace, Ashanti, Ghana | Dark green | 36.42 |
| 13 | AGBELIFIA | Locally released variety | Purplish green | 29.75 |
| 14 | BEDIAKO | Locally released variety | Light green | 39.24 |
| 15 | AGRA | Locally released variety | Purplish green | 32.19 |
| 16 | UCC-01-157 | Local, Landrace, Central Ghana | Dark green | 38.65 |
| 17 | Doku duade | Locally released variety | Dark green | 39.74 |
| 18 | Abrabopa | Locally released variety | Purple | 39.36 |
| 19 | KSI-00-179 | Local, Landrace, Ashanti, Ghana | Dark green | 44.24 |
| 20 | Essambankye | Locally released variety | Light green | 37.21 |
| 21 | IBA950289 | IITA | Purplish green | 34.66 |
| 22 | 1070557 | IITA | Purplish green | 30.29 |
| 23 | UCC-01-195 | Local, Landrace, Central Ghana | Purple | 33.15 |
| 24 | Broni | Locally released variety | Purplish green | 36.85 |
| 25 | TEKBANKYE | Locally released variety | Purplish green | 38.88 |
| 26 | IBA96/0581 | IITA | Purplish green | 32.98 |
| 27 | IBA98/0505 | IITA | Dark green | 30.88 |
| 28 | 1011797 | IITA | Dark green | 37.35 |
| 29 | TA-97-047 | Local, Landrace, Brong Ahafo, Ghana | Dark green | 40.13 |
| 30 | Amasen | Locally released variety | Purplish green | 39.22 |
| 31 | KSI-00-001 | Local, Landrace, Ashanti, Ghana | Purplish green | 39.38 |
| 32 | 1082264 | IITA | Dark green | 29.1 |
| 33 | IBA9102324 | IITA | Purplish green | 34.00 |
| 34 | UCC BANKYE | Local, Landrace, Central Ghana | Dark green | 35.32 |
| 35 | IBA011368 | IITA | Purplish green | 33.25 |
| 36 | IBA020431 | IITA | Light green | 39.36 |
| 37 | IBA993073 | IITA | Purplish green | 34.06 |
| 38 | Duade kpakpa | Locally released variety | Purple | 39.95 |
| 39 | SW-00-006 | Local, Landrace, Ashanti, Ghana | Dark green | 41.77 |
| 40 | UCC-00-002 | Local, Landrace, Central Ghana | Purplish green | 37.15 |
| 41 | Ampong | Locally released variety | Dark green | 37.8 |
| 42 | TME 149 | IITA | Dark green | 23.06 |
| 43 | Fil-Indiakonia | Locally released variety | Purplish green | 36.63 |
| 44 | UCC-01-004 | Local, Landrace, Central Ghana | Dark green | 43.54 |
| 45 | ADE-00-097 | Local, Landrace, Ashanti, Ghana | Purple | 27.74 |
| 46 | Bankyehemaa | Locally released variety | Purplish green | 40.83 |
| 47 | OTUHIA | Locally released variety | Purplish green | 21.58 |

*(Continued)*

**Table 1.** (Continued)

| Accession Number | Name | Source/region of collection | Colour of apical leaves | Average % roots dry matter |
|---|---|---|---|---|
| 48 | UCC-01-296 | Local, Landrace, Central Ghana | Dark green | 41.74 |
| 49 | DUDZI | Locally released variety | Purplish green | 34.14 |
| 50 | O.D | Local, Landrace, Ashanti, Ghana | Dark green | 38.05 |
| 51 | 1083724 | IITA | Purplish green | 35.58 |
| 52 | BD-96-157 | Local, Landrace, Eastern, Ghana | Purplish green | 38.56 |
| 53 | IBA 011371 | IITA | Purplish green | 35.68 |
| 54 | UCC-01-243 | Local, Landrace, Central Ghana | Dark green | 34.04 |
| 55 | AFS-00-050 | Local, Landrace, Central Ghana | Dark green | 40.77 |
| 56 | UCC-01-184 | Local, Landrace, Central Ghana | Purplish green | 30.91 |
| 57 | 1090151 | IITA | Light green | 29.36 |
| 58 | Debor | Local, Landrace, Ashanti, Ghana | Dark green | 35.15 |
| 59 | BD-96-115 | Local, Landrace, Eastern, Ghana | Purplish green | 40.55 |
| 60 | BIABESE | Local, Landrace, Tamale, Northern Ghana | Purplish green | 32.46 |
| 61 | OFF-00-023 | Local, Landrace, Ashanti, Ghana | Dark green | 39.8 |
| 62 | Nyerikogba | Locally released variety | Purplish green | 36.71 |
| 63 | 96/1613 | IITA | Purplish green | 40.27 |
| 64 | IBA010034 | IITA | Dark green | 37.03 |
| 65 | 01/1206 | IITA | Light green | 31.97 |
| 66 | IBA30572 | IITA | Purplish green | 35.73 |
| 67 | SW-00-220 | Local, Landrace, Ashanti, Ghana | Purplish green | 42.22 |
| 68 | ADE-00-046 | Local, Landrace, Ashanti, Ghana | Light green | 40.25 |
| 69 | WCH-00-037 | Local, Landrace, Brong Ahafo, Ghana | Purplish green | 38.29 |
| 70 | Afisiafi | Locally released variety | Purplish green | 38.81 |
| 71 | UCC-01-249 | Local, Landrace, Central Ghana | Dark green | 42.32 |
| 72 | ADE-00-038 | Local, Landrace, Ashanti, Ghana | Dark green | 42.94 |
| 73 | 1083703 | IITA | Purplish green | 36.1 |
| 74 | BD-96-141 | Local, Landrace, Eastern, Ghana | Dark green | 43.75 |
| 75 | Eskamaye | Locally released variety | Purplish green | 40.33 |
| 76 | 1083774 | IITA | Purplish green | 38.15 |
| 77 | ABASAFITA | Locally released variety | Dark green | 41.11 |
| 78 | OFF-00-087 | Local, Landrace, Ashanti, Ghana | Dark green | 36.85 |
| 79 | NKABOM | Locally released variety | Dark green | 36.8 |
| 80 | KSI-00-036 | Local, Landrace, Ashanti, Ghana | Dark green | 38.88 |
| 81 | IBA061635 | IITA | Purplish green | 37.1 |
| 82 | KW-00-148 | Local, Landrace, Ashanti, Ghana | Dark green | 40.8 |
| 83 | UCC-01-461 | Local, Landrace, Central Ghana | Purple | 34.99 |
| 84 | NKZ-00-034 | Local, Landrace, Brong Ahafo, Ghana | Purple | 42.29 |
| 85 | UCC-01-218 | Local, Landrace, Central Ghana | Dark green | 41.21 |
| 86 | 1085392 | IITA | Light green | 34.87 |
| 87 | UCC-01-011 | Local, Landrace, Central Ghana | Dark green | 42.49 |

**DArT procedure.** The DArT arrays were produced from libraries prepared from *PstI*-based genomic representations [38]. The genomic representations were generated by digesting 100 ng mixtures of DNA samples with 2 U *PstI* and a frequent cutter (*BstNI or TaqI*) (NEB) in a buffer containing 10 mM Tris-OAc, 50 mM KOAc, 10 mM Mg(OAc)$_2$ and 5mM DTT as suggested by Xia et al. [38, 46, 47] for genome complexity reduction. Fragments were sequenced on HiSeq 2500 (Illumina). Libraries were sequenced from one end by performing

single read sequencing runs [38]. The SNP markers were searched and filtered using algorithms. The sequenced data were analyzed using DarTsoft14, an automated genotypic data analysis program and DArTdb, a laboratory management system. Markers were scored '1' for presence, and '0' for absence and '-' for calls with non-zero count but too low counts to score confidently as "1" for the SilicoDArT while the DArTSeq SNPs were scored '0' for reference allele homozygote, '1' for SNP allele homozygote and '2' for heterozygote.

**Marker quality parameters.** Both marker systems were tested for their PIC, reproducibility (%) and call rate (%). PIC indicates the diversity of the marker in the population and showed the usefulness of the marker for linkage analysis while reproducibility involved the proportion of technical replicate assay pairs for which the marker score exhibited consistency [48]. Call rate (%) was also used to eliminate markers with ≥5% missing data.

**Genetic relationship analysis.** Genetic relationships among the accessions were estimated based on Gower's dissimilarity index for the set of DArT markers [49]. Accessions were declared potential duplicates if the dissimilarity between them fell within the threshold of the replicated DNAs. The "pvclust" package was used to generate the Gower's dissimilarity matrix and "cluster" package in R was used to construct average linkage hierarchical dendrogram for SilicoDArT and DArTSeq SNPs data [50, 51]. Correlation between both marker systems was determined using the Mantel test as implemented in the "vegan" package of statistical software program 'R' by employing 10,000 random iterations in the non-parametric test calculator while the "ggplot2" package was used to generate the Mantel test scatterplot [52, 53].

**Genetic diversity and population structure analysis.** STRUCTURE v.2.3.4 [54] was used to analyse the genetic structure of the initial 87 cassava collection and 67 individuals upon roguing off the potential duplicates. The number of hypothetical subpopulations (K) was estimated with the STRUCTURE software through the application of a Bayesian clustering approach for the organisation of genetically similar cultivars into the same subgroups. Admixture and shared allele frequencies model was used to determine the number of clusters (K) in the range from 1 to 10. For each run, the initial burn-in period was set to 20,000 followed by 30,000 MCMC (Markov chain Monte Carlo) iterations, with no prior information on the origin of individuals. Longer burn-in or MCMC has been reported not to change significantly the results [55]. The ΔK method was used to determine the most suitable value of K as implemented in Structure Harvester [55]. The structure results for the assumed population (1–10) were subsequently analysed online using the STRUCTURE HARVESTER [56] to identify a distinct peak of the curve in the change of likelihood (ΔK) at the true value of K. Principal Coordinate Analysis (PCoA) of the DArTseq markers was performed using PAST software v.3.14 [57].

## Results

### Quality parameters of markers

A total of 31865 SNP and 32377 SilicoDArT markers were identified upon application of the complexity reduction method with a call rate in the range of 81–100% and 70–100% respectively (S1 Fig). Around 22516 SNPs and 27182 SilicoDArT markers were assigned to 18 haploid chromosomes of cassava after aligning to cassava_v61 and v8 model reference. These ranged from 910 markers on chromosome18 (chr18) to 2158 on chromosome1 (Fig 1) for the dominant SilicoDArT markers. A total of 10521 SilicoDArT markers (S1 Table) passed the quality parameters with 100% reproducibility, and 97.4% mean call rate. The selected 10521 markers were very informative with a PIC range of 0.2–0.5 and an average of 0.36 (Fig 2).

Also, 10808 SNPs (S2 Table) with 100% reproducibility and call rate passed the quality test and were selected for further analysis. The mean PIC of these selected markers was 0.28 which

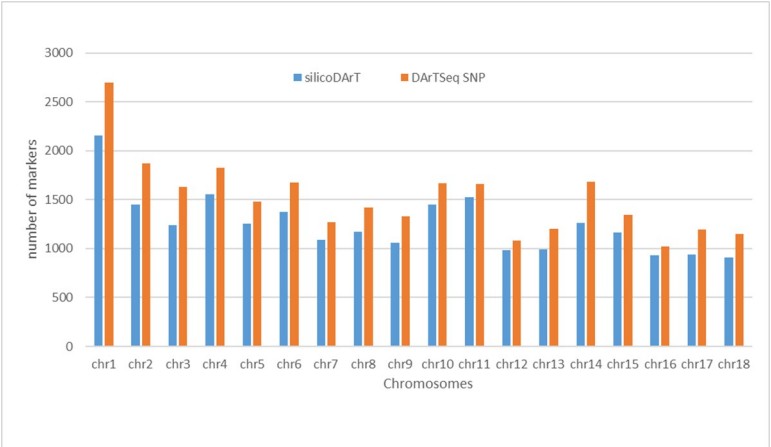

**Fig 1. Genomic distribution of markers across the chromosome of cassava.**

was relatively lower than that of the SilicoDArT markers. Around 20.65% of the SNPs were in the lowest PIC value range of 0<0.10 while 32.42% were in the highest range of 0.4–0.5 (Fig 2). A genome-wide SNP density plot revealed that 1019 to 2696 SNPs were physically mapped to chromosome16 (chr16) and chromosome (chr1) respectively (Fig 1).

Analysis of the type of SNP (**Table** 2) in the selected SNPs revealed that transition SNPs (50.60%) were closely similar to transversions (49.40%). Among the six SNP types (**S2 Table**), A/G transitions (0.256) had the highest frequency though it was similar to C/T transitions (0.250) while G/C transversions were the least (0.101).

## Genetic relationships among the cassava accessions

Genetic relationships were estimated among the accessions through their genetic distances using the selected SilicoDArT and SNP data based on Gower's genetic dissimilarity.

The overall genetic distance ranged from 0.00 to 0.41 with an average of 0.30 as revealed by the DArTSeq SNPs (**S3 Table**). The range of genetic dissimilarity among the IITA lines was between 0.01–0.40 with a mean of 0.33 which was similar to that of the improved ones (0.34). The lowest average genetic distance was found within the landraces (0.20). The critical distance threshold to declare whether two genotypes are duplicates/clones was empirically determined

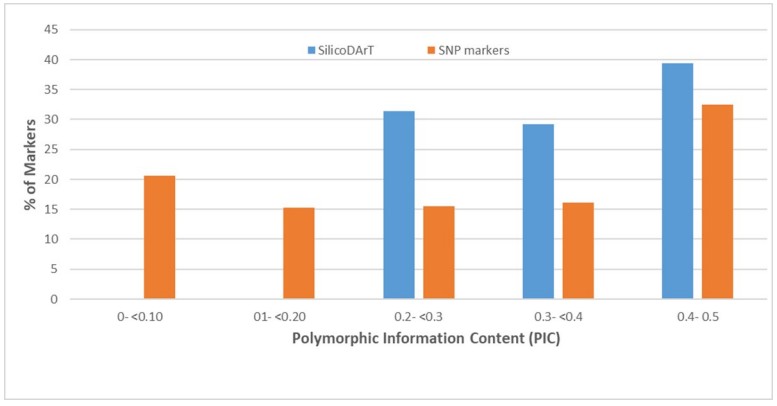

**Fig 2. Percentage distribution of PIC values of selected SilicoDArT and DArTSeq markers.**

**Table 2. Distribution of transition and transversion SNPs across the cassava genome.**

| Type of SNP | Transitions | | Transversions | | | |
|---|---|---|---|---|---|---|
| | A/G | C/T | AC | AT | GC | GT |
| Number of allelic sites | 2764 | 2706 | 1379 | 1576 | 1090 | 1293 |
| Individual frequencies | 0.256 | 0.250 | 0.128 | 0.145 | 0.101 | 0.120 |
| Total | 5470 (0.506) | | 5338 (0.494) | | | |

from the distribution of pairwise distances between replicated DNAs from three samples. The genetic dissimilarity index between the duplicated DNAs fell within the range of 0.00–0.01 and 0.00–0.02 for the SNPs and SilicoDArT respectively hence accessions with pairwise distances within these ranges were referred to as being potentially redundant. Therefore 22 landraces (UCC-BANKYE, O.D: DMA-2000-024, UCC-2001-015, BD-96-065, KSI-2000-191, UCC-2001-157, KSI-2000-179, TA-97-047, UCC-2001-004, UCC-2001-296, UCC-2001-243, AFS-2000-050, Debor, OFF-2000-023, UCC-2001-249, ADE-2000-038, OFF-2000-087, KSI-2000-036, KW-2000-148, UCC-2001-218, UCC-2001-011) constituting 53% of the total landraces obtained from the genebank were found to be potential duplicates. Debor, a known landrace was found to be similar to 19 other landraces collected from different regions. The IITA lines were generally closely related to the improved varieties than the landraces. Grouping of the accessions based on average linkage hierarchical clustering gave two main clusters (C1 and C2) containing related cassava accessions with common origin or shared parental lines (Fig 3). Several of the IITA lines (64%) predominantly grouped in C2 (green) while the landraces (90%) grouped in C1 (red). The improved varieties were evenly distributed among the two clusters.

After employing the SilicoDArT data, the genetic distance among the 87 accessions ranged between 0.0 to 0.61 (S4 Table). The average genetic dissimilarity of 0.38 among the accessions revealed by the SilicoDArT was higher than that of the SNP markers (0.30). Also, a low mean genetic dissimilarity of 0.24 was found among the landraces. Again, the 22 landraces with pairwise distance within the 0.00–0.02 range or threshold were found to be redundant confirming what was revealed by the SNPs. Ten out of these 22 came from one region (UCC, central region, Table 1) manifesting the sharing of plant material among farmers within the vicinity. Generally, the genetic distance between the groups of accessions was higher with the SilicoDArT than that observed through the SNP markers. The dendrogram obtained with the

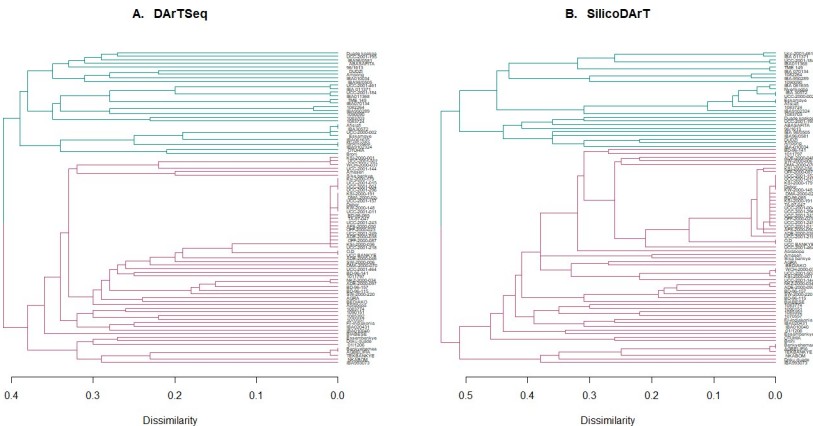

**Fig 3. Average linkage hierarchical dendrogram showing genetic relationship among 87 cassava genotypes (C1 = red, C2 = green).** A) Based on DArTSeq SNPs. B) Based on SilicoDArT markers.

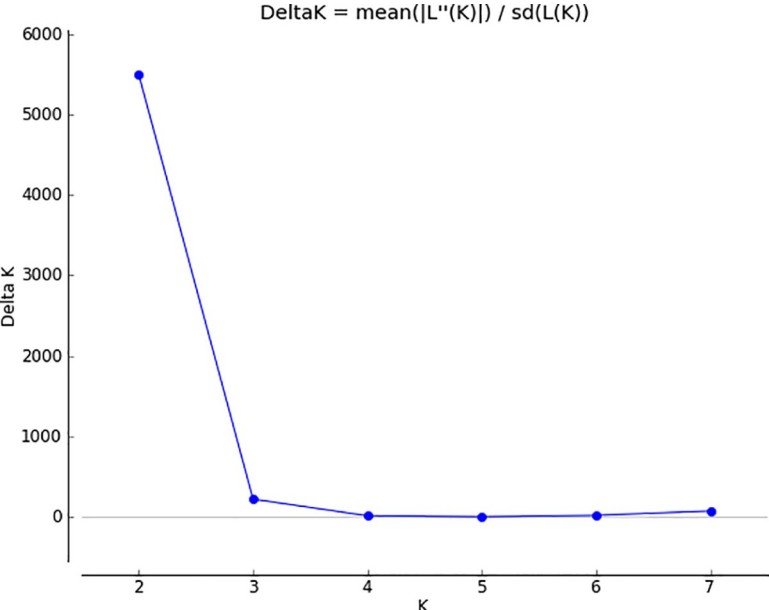

**Fig 4. Delta K for different numbers of subpopulations (K).**

SilicoDArT markers produced two main clusters (C1 and C2) and its adjoining subclusters (**Fig 3**). The IITA lines mainly grouped in C2 (green) while the landraces (90%) grouped in C1 (red) just as was observed with the SNP markers.

## Diversity and population structure based on DArTSeq SNP markers

The SNP markers were used for estimating the genetic structure of the cassava population using the Bayesian clustering model implemented in the computer software STRUCTURE. The simulations (logarithm probability relative to standard deviation, ΔK) estimated from the SNP markers showed a sharp peak at K = 2 (**Fig 4**) which is the real K similar to what was observed by Hampton et al. [58]. This means that the optimum number of subpopulations is two.

The population assignment test with the total 87 samples in the structure analysis shows the overall proportion of membership of the sample in each of the two clusters as illustrated in the bar plot for K = 2 (Fig 5). The two subpopulations (subpop1 and subpop2) consisted of 35.4% and 64.6% members respectively (Table 3). The hypothetical founder population seen in sub-pop1 (red) is represented by landraces while that of subpop2 (green) consisted mainly of IITA

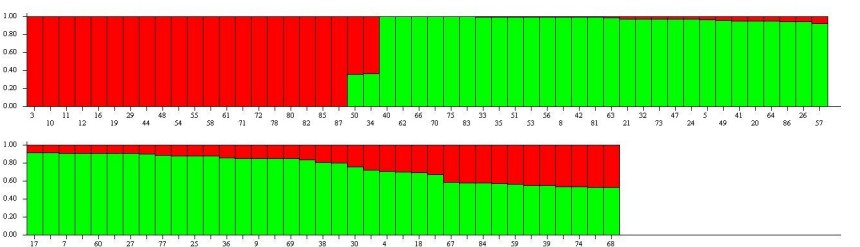

**Fig 5. Estimated population structure of 87 cassava accessions on K = 2.** Accessions in red were clustered into subpop1 and the ones in green were clustered into subpop2.

and improved lines similar to what was revealed by the average linkage hierarchical clustering. Twenty accessions which were mainly landraces from Ghana were fully assigned in subpop1. These accessions were evident as C1 in the dendrogram and were the least divergent accessions based on Gower's dissimilarity. In addition, these 20 landraces were fully conserved in one group even when higher K (K = 9) was assumed (S2 Fig) indicating these accessions could be clones. On other hand, 15 lines comprising of nine IITA lines (IBA070134, IBA9102324, IBA011368, TME 149, 1083724, IBA 011371, IBA30572, IBA061635, 96/1613), three improved (Nyerikogba, Afisiafi, Eskamaye) and three putative landraces (UCC-2000-002, UCC-2001-184, UCC-2001-461) were also entirely assigned to subpop2. About 52 accessions showed admixtures (with ≥1% of ancestry from subpop1 or subpop2) of subpop1 and subpop2 genetic composition. Majority of the IITA lines (68%) were found to be in admixture. These results conformed with the average linkage hierarchical clustering developed through the set of marker systems (Fig 3).

The expected heterozygosity from the STRUCTURE analysis was used to define the average genetic diversity between individuals within each subpopulation (subpop). The expected heterozygosity (Table 3) of subpop1 was 0.008 while that of subpop2 was 0.391 indicating the homogenous and the diverse nature of subpop1 and subpop2 respectively. Divergence among the two subpopulations shown by Net nucleotide distance (0.22) revealed that subpop1 was moderately distantly related to subpop2 **(Table 3).**

Further population structure analysis was performed among 67 accessions upon purging the potential duplicates. The optimum number of subpopulations was again found to be two mainly made of landraces in SUBPOP1 and IITA and improved in SUBPOP2 while 45 accessions were in admixture (S3 Fig). The expected heterozygosity increased from 0.008 to 0.357 for subpop1 while that of subpop2 remained almost the same (0.368) (Table 3)

A principal coordinate analysis (PCoA) based on the pairwise Gower's genetic distance matrix among the accessions was performed to depict the genetic divergence in the cassava lines using the two variants of DArT markers. Using the SNP markers, 38.2% of the total genetic variation was explained by the first two axes of the PCoA (Fig 6A). The first two axes of the PCoA based on the SilicoDArT (Fig 6B) explained 47.2% of the total genetic divergence. The distribution of the accessions based on the two marker systems was similar which was also consistent with the average linkage hierarchical clustering (Fig 3) and the structure analysis (Fig 5). The landraces (black) clustered together while the IITA lines (green) also clustered together. Though the improved varieties (blue) showed wide diversity, they were predominantly clustered with the IITA lines.

## Association among the two DArT markers systems

Comparisons of the relationship among the accessions based on Gower's distance matrices derived from the SilicoDArT and DArTseq SNP markers depicted high association (r = 0.970; P < 0.001) between both markers systems through the Mantel correlation test (S4 **Fig**). The result showed a good fit between SilicoDArT and DArTseq SNP markers data sets in assessing genetic diversity in the cassava germplasm.

**Table 3. Net nucleotide distance (diversity between populations) and expected heterozygosity (average diversity within populations) and proportion of membership in each subpopulation.**

| Subpopulations | Net Nucleotide Distance | Expected heterozygosity | Proportion of Membership | Expected Heterozygosity of 67 samples |
|---|---|---|---|---|
| | subpop2 | of 87 samples | (87 samples) | |
| subpop1 | 0.22 | 0.008 | 0.354 | 0.357 |
| subpop2 | - | 0.391 | 0.646 | 0.368 |

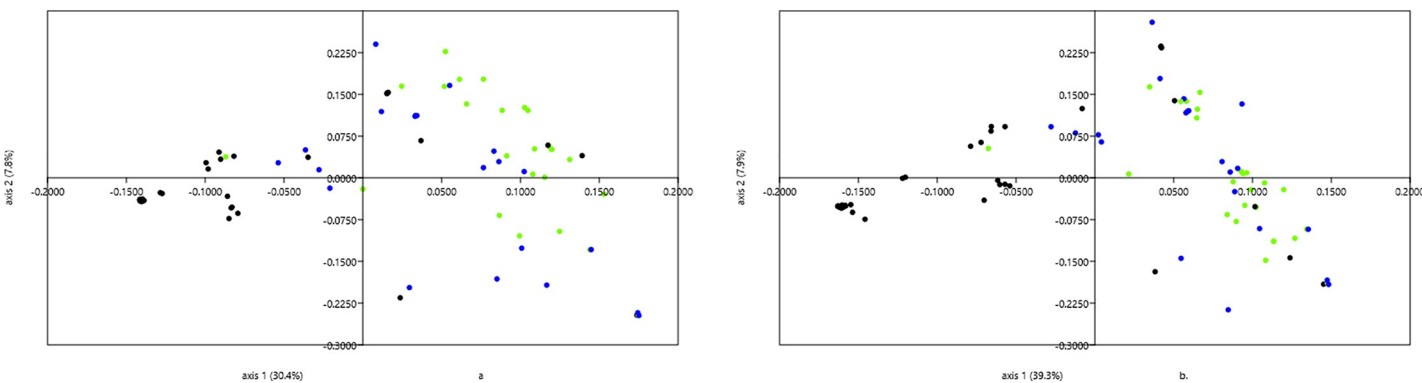

**Fig 6.** Principal coordinate analysis of cassava accessions (landraces = black, improved = blue, green = IITA) (a) based DArTSeq SNPs (b) based SilicoDArT markers.

## Discussion

### Genome-wide marker discovery and quality analysis

Genome level profiling of crop germplasm collections is a critical initial step in the identification of duplicates and divergent parents for effective conservation and utilization in breeding programs. Cassava is known worldwide for domestic and industrial use however, its classification and conservation in germplasm banks is challenging, due to phenotypic plasticity of cassava being further complicated by its asexual life-cycle [25, 59]. This study highlights the potential of highly informative and selective SilicoDArT and DArTSeq SNP markers for genomic studies in cassava which might underpin conservation and future breeding efforts. A total of 10521 SilicoDArT markers and 10808 informative DArTSeq SNPs were used for genetic diversity studies and population structure analysis. Future cassava breeding programs depend on the usage of a large number of these high-throughput markers for effective selection and genome association studies. The quality parameters of the selected markers were comparable with that of others plant species. The mean PIC of 0.36 for the dominant SilicoDArT markers was lower than the 0.42 identified in cassava and its wild relatives by Xia *et al.* [46] and 0.41 found in sorghum [60] but higher than the 0.28 in *Beta vulgaris* [61] and 0.29 in macadamia [48]. The average PIC of the SNPs (0.28) on the other hand was higher than the 0.228 identified in cassava [35], 0.21 in macadamia and 0.265 in Durum wheat [62]. The average PIC value of SilicoDArT was greater than that of SNP markers similar to that reported by Alam et al. [48] in Macadamia indicating the SilicoDArT were more informative than the SNP markers. The use of these high-density SilicoDArT and SNP markers may achieve better genome coverage through the sampling of a greater number of points in the whole genome, as marker density is reported to have a high correlation with gene density [38, 63]. Earlier reports on diversity studies on cassava utilized relatively smaller number of molecular markers; 35 SSR [33], 26 SNPs [35] and 4 RAPD [64], hence these high-density SilicoDArT and SNP markers may better suit for robust genomic and conservation activities in cassava. Additionally, the co-dominant inheritance pattern of SNP markers may increase the utility of DArT platforms for genetic population analysis. Relative to other marker technologies, DArT markers are suitable for high-throughput work as well as being cost effective [46, 65].

Similar to previous studies with relatively fewer SNPs within genes involved in cyanogenesis (CYP79D2), starch metabolism (GBSSII) and defense pathways within cassava, transition SNPs were similar to transversions with the most frequent transition and transversion being A/G and A/T, respectively [35]. Higher DNA polymorphisms are expected from out-crossing and inbreeding-sensitive crops like cassava partly due to the inherently high number of loci

maintained in a heterozygous state. This contradicts what was reported in crops like *Camelina sativa* [12], rubber tree [66], *Brassica napus* [67] where transitional SNPs substantially exceeded transversion SNPs.

## Genetic relationship among accessions

The assessment of genetic diversity is a vital pre-breeding activity towards crop improvement and efficient conservation of crop biodiversity. A genetic distance approach based on Silico-DArT and DArTSeq SNPs were successfully used to ascertain the relationship among the accessions as well as revealing potential duplicates. Similar to an earlier report by Alam et al. [48], the average genetic distance among the cassava accessions was higher with the dominant SilicoDArT than the SNPs (**S3 and S4 Tables**). Two major clusters were formed using both marker systems. The dendrogram (**Fig 3**) created using the average linkage hierarchical clustering method for both sets of markers grouped the landraces into C1 (red) and the IITA lines in C2 (green) showing their relationship with their pedigrees.

Base on the markers, the highest average dissimilarity index was found among the improved lines which were uniformly distributed among the two clusters probably due to their diverse pedigree or origin. Ferguson et al. [68] found elite lines from ESC Africa and IITA breeding lines to be more closely related and indicated that, this could be as a result of the movement of germplasm from IITA to ESC Africa through collaborations. Similar to the situation in Ghana, the known improved cassava varieties documented in the *Catalogue of crop varieties released and registered in Ghana* had their pedigree/line from IITA and/or the local landraces so it was not surprising the improved lines were evenly distributed among the two clusters [69]. Cassava, unlike crops such as maize, landraces are often indistinguishable from improved clones and are often considered for release hence, breeding has not significantly separated improved clones from landraces [68].

The landraces on the other hand had the lowest mean dissimilarity index due to the high level of duplicates that were detected within the group. Both marker systems identified 22 landraces to be within the distance threshold of the replicated DNAs. These were identified as potential duplicates. The lack of a formal seed system for cassava production promotes the sharing of planting materials among farmers as most farmers use their own planting materials (usually stem cuttings from the preceding crop) or they source stem cuttings from neighboring farmers especially for varieties with good culinary attributes for cultivation [70, 71]. These result in the renaming of the accessions leading to increased synonymy and homonymy which may confound the true diversity within the accessions when relying on the use of local/vernacular names alone [70, 72, 73]. Most of these landraces classified as potential duplicates were independently collected from different regions of Ghana and therefore came with different accession names. Though morphologically characterized, the low resolution of morphological markers could not reveal the redundant accessions [25, 59, 74]. Debor has excellent culinary traits, two of which are mealiness after boiling and relatively sweet taste hence it was not surprising it was found to be redundant with 19 landraces. This aligns with the report by Rabbi et al. [71] between Debor and other cassava genotypes. The complementary results by the SilicoDArT and SNP markers provide enough evidence to cull/rogue off redundant lines for efficient conservation and subsequent breeding.

## Structure of the population

Population structure analysis provides helpful information in maintaining and monitoring the genetic diversity required for a robust breeding program [75]. Results of population structure analysis among the original 87 samples revealed only two major subpopulations (**Fig 5**)

analogous to the report by Adjabeng-Danquah et al. [33]. They studied some of these accessions with 35 SSR markers but indicated a clear genetic divergence based on the origin of the cassava genotypes. The grouping of the landraces based on the Bayesian model showed similar results as reflected in average linkage hierarchical clustering (**Fig 3**) and principal coordinate analysis. The genetic structure present in the population meets our expectations based on the sources/pedigree of the genotypes. All the genotypes were originally collected from two different locations (1. Landraces + improved = local, 2. IITA). Subpop1 consisted of only landraces while the IITA and other genotypes were clustered in subpop2.

In addition to these distinct two subpopulations, a large number of admixtures was obtained among the cassava population studied [12, 33]. The expected heterozygosity was used to express the genetic diversity between individuals within each subpopulation (**Table 3**). The expected heterozygosity of subpop1 was 0.008 which was very low indicating the large homogeneity of the individuals in the subgroup. This was also seen in the low level of admixture in subpop1 as revealed by the STRUCTURE results (**Fig 5**). As evidenced in the cluster analysis, the 22 landraces found to be redundant based on Gower's dissimilarity constituted subpop1 hence the low genetic diversity between the genotypes in the subgroup was not surprising. This shows a true reflection of the exchange of plant materials with good economical and culinary traits among farmers within the country resulting in the generation of such duplicates and the high resolution of molecular markers over morphological ones [59, 76, 77]. In contrast to subpop1, the average diversity between individuals within subpop2 was 0.391 which shows the significant divergence within the subgroup (**Table 3**). This level of differentiation of individuals inside the group indicates that even among individuals clustered by genetic proximity, there is still high variability [78]. The high level of admixture found in subpop2 again reflects the genetic variation within the group [62]. The high level of admixture found in the IITA lines in subpop2 corroborates what was reported by Adjabeng-Danquah et al. [33]. Also, the diverse background (source) of the accessions in subpop2 relative to subpop1 could influence the amount of variation between individuals in the subpopulation. The collections obtained from IITA could likely be made up of accessions originating from various geographical areas in West Africa [68]. As witnessed in most of the released improved cassava varieties in Ghana, they had their parental sources from IITA, indicating the formal exchange of breeding materials across borders [69] which could lead to 'secondary' mixing of the gene pool of the collections through hybridization and enhanced variation.

The variation between the two subpopulations defined by the Net nucleotide distance (Allele-frequency divergence among populations, 0.22) showed a moderate genetic differentiation between subpop1 and subpop2 (**Table 3**). Therefore, the genetic variation in the cassava germplasm is captured within and between subpopulations while a high level of similarity also exists within the subpopulation (subpop1). Adjabeng-Danquah et al. [33] reported a high level of variation within cassava groups similar to what was found here.

Further population structure analysis without the potential duplicates gave a higher average diversity within subpop1 (expected heterozygosity, 0.357 for SUBPOP1 and 0.368 for SUBPOP2) which was similar to the 0.333 average reported by Ferguson et al. [68] within groups. This supports the earlier revelation of a high level of duplicates among the landraces resulting in low expected heterozygosity in subpop1 and the need to purge them. This seems to be a common phenomenon among gene banks since the earlier characterizations of collections were based on agro-morphology which are phenotypically plastic or limited DNA-based markers [24, 25]. Similarly, a recent study found several duplicates within the gene bank at IITA, and efforts are been made to purge these duplicates [68]. This information is necessary particularly for the selection of individuals to serve as parents in breeding programs.

The set of DArT platforms were very informative in revealing the genetic variations in the germplasm as earlier reported [38, 48]. This was confirmed by the high correlation index revealed by the Mantel test that was conducted to check the association between both marker systems (S3 **Fig**). The consistency in genetic relationships among the cassava by the Silico-DArT and SNPs suggest that the two DArT marker systems are highly reliable for genetic diversity study in cassava. The available population structure is a step towards efficient germ-plasm conservation and parental selection to take advantage of heterosis breeding.

## Conclusion

In this study, high-density dominant SilicoDArT and codominant DArTSeq SNP markers were used to explore genetic diversity and population structure among a collection of cassava. Both DArT markers successfully revealed the parental relationships and the extent of diversity in the population by showing some degree of genetic diversity and duplicates in the collection. There was a high level of redundancy in the local landraces (53%) compared to those obtained from IITA. This level of genetic diversity could be the basis for developing new cassava cultivars with desirable characteristics while improving the *ex situ* cassava germplasm conservation activities at the gene bank. In addition, our study identified two subpopulations that could be explained by the pedigrees of the genotypes. The Structure analysis indicated subpop2 to be more diverse than subpop1 (0.008) based on expected heterozygosity. Further analysis following the removal of the potential duplicates increased the expected heterozygosity from 0.008 to 0.357. Both DArT platforms seem an inexpensive and robust option for genomics studies in cassava. The large number of highly polymorphic markers developed and the knowledge of population struc-ture and genetic diversity of cassava accessions will be important for cassava germplasm cura-tion, heterosis breeding and future marker-traits association studies and genomic selection.

## Supporting information

**S1 Fig. Call rate distribution of the two marker system.**
(TIF)

**S2 Fig. Population structure bar plot of 87 cassava accessions on K = 9.**
(TIF)

**S3 Fig. Population structure bar plot of 67 cassava accessions on K = 2.**
(TIF)

**S4 Fig. Mantel correlation test between SilicoDArT and SNP.**
(TIF)

**S1 Table. List of silicoDArT markers.**
(XLSX)

**S2 Table. List of DArT-based SNP markers.**
(XLSX)

**S3 Table. Gower's dissimilarity index among the cassava accessions based on SNP markers using the "cluster" package in R software.**
(CSV)

**S4 Table. Gower's dissimilarity index among the cassava accessions based on SilicoDArT markers using the "cluster" package in R software.**
(CSV)

## Acknowledgments

The authors thank Fuseini Mohammed and Edwin Appiah Moses in the extraction of DNA, data management and the anonymous reviewers for their insightful suggestions. The support from Alliance for Green Revolution Africa (AGRA) through the IMCDA programme at KNUST (2018) is also highly acknowledged.

## Author Contributions

**Conceptualization:** Bright Gyamfi Adu.

**Data curation:** Bright Gyamfi Adu, Stephen Amoah, Alex Yeboah, Richard Adu Amoah, Eva Gyamfuaa Owusu.

**Formal analysis:** Bright Gyamfi Adu, Daniel Nyadanu, Richard Adu Amoah, Eva Gyamfuaa Owusu.

**Funding acquisition:** Richard Akromah, Stephen Amoah.

**Investigation:** Bright Gyamfi Adu.

**Methodology:** Bright Gyamfi Adu, Richard Akromah, Stephen Amoah, Eva Gyamfuaa Owusu.

**Project administration:** Richard Akromah.

**Resources:** Lawrence Missah Aboagye.

**Software:** Bright Gyamfi Adu, Alex Yeboah, Richard Adu Amoah, Eva Gyamfuaa Owusu.

**Supervision:** Richard Akromah, Stephen Amoah.

**Validation:** Richard Akromah, Stephen Amoah, Lawrence Missah Aboagye.

**Visualization:** Stephen Amoah, Daniel Nyadanu.

**Writing – original draft:** Bright Gyamfi Adu, Lawrence Missah Aboagye.

**Writing – review & editing:** Bright Gyamfi Adu, Stephen Amoah, Daniel Nyadanu, Alex Yeboah.

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
