## [Decision Letter · Decision Letter 0]

8 Apr 2021

PONE-D-21-06686

High-density DArT-based silicoDArT and SNP markers for genetic diversity and population structure studies in cassava (Manihot esculenta Crantz)

PLOS ONE

Dear Dr. Adu,

Thank you for submitting your manuscript to PLOS ONE. After careful consideration, we feel that it has merit but does not fully meet PLOS ONE’s publication criteria as it currently stands. Therefore, we invite you to submit a revised version of the manuscript that addresses the points raised during the review process.

We look forward to receiving your revised manuscript.

Kind regards,

Tzen-Yuh Chiang

Academic Editor

PLOS ONE

Journal Requirements:

2. Please include a caption for figures 4-8.

Reviewers' comments:

Reviewer's Responses to Questions

**Comments to the Author**

1. Is the manuscript technically sound, and do the data support the conclusions?

Reviewer #1: Partly

Reviewer #2: Partly

2. Has the statistical analysis been performed appropriately and rigorously? 

Reviewer #1: Yes

Reviewer #2: Yes

3. Have the authors made all data underlying the findings in their manuscript fully available?

Reviewer #1: No

Reviewer #2: Yes

4. Is the manuscript presented in an intelligible fashion and written in standard English?

Reviewer #1: No

Reviewer #2: No

5. Review Comments to the Author

Reviewer #1: High-density DArT-based silicoDArT and SNP markers for genetic diversity and population structure studies in cassava (Manihot esculenta Crantz)

Header: Take out additional symbols after number superscript

Abstract:

Ln 1: Take out extra spaces throughout the document; just search for space space; Manihot esculenta italix

You need to say somewhere what the germplasm is? Landraces, breeding lines and from where? Before you get into results

Introduction:

Ln 52 due to its ability

Ln 53 Take out “than most staple crops”

Ln 54: contain

Ln56:]

Ln 58: Okogbenin and FrEgene – this work was a long time ago, and much has happened since then. Cassava is no longer an ‘orphan’ crop.

Ln 102: DArTSeq does not involve microarray. It purely involves complexity reduction (restriction digestion), and NGS. The authors need to be aware that before NGS came about, the company used a microarray based genotyping technique called DArT. This is not used much now, if at all. It seems the authors have got confused with the two techniques.

Ln 106: I understand that SilicaDArT are a reflection of epigenetic status. They are epigenetic markers. The difference between DArTseq SNPs and Silico needs to be VERY clearly defined. Microarray is a technology, not a form of marker. It is not used in generating SilicoDArT.

Ln107: Be consistent in how you define DArT-based SNP or DArTseq SNP.

Ln 109/110: With DarT there is no choice in restriction fragments.

Ln 114: Be more specific about the genebank

Materials and methods

Ln120 eighty-seven should be 87

Ln 121: How many were landraces and how many released cultivars?

Ln 124: twenty-five should be 25

Ln 127: This is the fist time you mention morphological. This should be mentioned in introduction.

Table 1: Why choose to report colour of apical leaves? Why dry matter? Also, cassava has storage roots, not tubers.

DArT procedure paragraph: Please be aware that DArT and DArTSeq are different procedures. There is no cloning in DArTSeq. This paragraph needs to be corrected.

Ln 164: You need to be clear that alleles were scored as 1 and 0; provided in two line format; or 0,1 and 2 as single row format with heterozygotes identified. This is critical. Distinguish scoring between Silico DArT and DArtSeq.

Ln 175: Don’t use DArT-based SNPs – these are DArTSeq SNPs

Ln179: I would use the term duplicates rather than redundant.

Ln 178: You need to discuss why you used Gowers

Ln 180: “pvclust” in R (include reference)

Ln 182: SilicoDArT and DArTSeq SNPs.

Ln 198: Why capital letters? I wouldn’t call it a peak; rather a flattening of the curve.

Results:

Ln 207: What do you mean “18 haploid cassava genome”? Explain this. You mean chromosomes?

Fig 1a and b. These figures are not necessary.

Fig. 2 needs to be better quality

Fig 3. Axis labels should not be capitals. Bars must not overlap and needs better quality.

Fig. 4. Cannot read anything from this Figure.

None of the subsequent figures have numbers.

Ln 253 : Use SilicoDArT throughout the document.

Ln 271: As far as I know you are looking for a flattening of the graph where deltaK changes, not a peak.

Ln 275: Please explain the Average Silhouette approach

Ln 276: You need to define which accessions are in which subpop before you start talking about diversity. Individuals are allocated to sub-populations.

Ln 292: Give a reference to average linkage clustering. What is meant by this. Clarfy.

Ln 302: Three rather than 3; and I would call these putative landraces as they may actually be improved lines

Ln 304: all cassava is heterogeneous. This is not a good word. Perhaps use Admixture.

Ln 308: Is there a figure associatd with this?

Ln 316: I wouldn’t indicate left of right as the figure could be switched. Just give colour. Rather say cluster together…

Ln 338: What do you mean 22,516 assigned to specific chromosomes?

Ln 341: I wouldn’t say these markers have to be developed. They are there, they just need to be visualized.

Ln 350: better genome coverage than what??

Ln 353: What about Ferguson et al. (2019) and many studies by Rabbi et al.

Ln 369: This is not two DArT platforms – just two types of data are generated form one platform

Ln 408: two

Ln 411: Rather than heterogeneous, I would say admixture or relatively close relationship.

Ln 416: The authors must understand the basic rules of reporting numbers; 1-10 should be written in full. Anything more than 10 is given in figures. Please correct throughout.

It does not indicate anywhere whether the duplicates were taken out of the dataset before diversity analysis was conducted. This should be done.

Ln 429: You need to find out the background / pedigree of IITA material

Ln 444: You might want to say that DArTSeq SNPs are preferred as they are co-dominant which means heterozygotes can be determined.

Conclusions

Ln 451: It is not an array that is used. Be consistent in naming the technologies throughout the doc.

Ln 459: I don’t think it is defined by geographical differentiation, but more by pedigrees – breeding lines vs landraces.

Reviewer #2: General comments:

The manuscript reports genetic diversity of cassava from Ghana. The study involves more than 80 accessions including improved varieties and landraces that were genotyping using a reduced-representation library sequencing (DArT). The authors presented standard diversity and population genetics analysis results including a report on marker informativeness, pairwise genetic distances among the accessions, hierarchical clustering, PCA and ancestry-based population structure.

Specific comments

1. While the study contributes to the body of knowledge about the diversity of the crop in Ghana, the authors repeatedly put forth statements that are rather outdated when justifying their study in the introduction section of the manuscript. For example, there is repeated mention that cassava is an orphan crop, with no genomic resources available and this is not correct (e.g. lines 28 - 30 and Lines 58 - 61). Indeed, one of the papers used to support this notion is almost two decades old (Okogbenin et al. Theoretical and Applied Genetics. 2003; 107:1452–1462.)

There are many high-quality reference genomes, hundreds of accessions have been whole-genome resequenced to produce hap-map, and several diversity analysis that used genotyping-by-sequencing have been published. Moreover, genomic resources have been used to map QTLs using biparental populations and GWAS. Moreover, genomic selection in cassava is quite advanced and comparable with other major crops. It is therefore not correct for the authors to say that there is very little progress in the crop. A rigorous literature review should address this misperception.

2. The way the manuscript is written is not concise and clear. There are many repetitions and the same thing are being presented multiple times.

Line 42 - 45: seems like a concluding statement that should have been at the end of the abstract.

L58 - 61: fragmented sentence that is not understandable. What is the intended message?

L80 - 81: "... as detailed by some authors" is not necessary. Similar types citation styles are in the manuscript. It should just be "The use of molecular tools in plant genetic analyses and crop improvement cannot be overemphasized [26-29]."

L84 - 92: The authors introduce a description of other types of molecular markers, including anonymous markers such as RFLPs etc. What purpose does this serve? Back when SNPs were new, it was acceptable to describe its advantages compared with the traditional markers but this serves no purpose since SNP have become mainstream.

Instead, the authors should describe DArTseq and compare it with other high-density SNP genotyping methods like GBS which has been extensively used for cassava.

L96-97: Why would one want to use "sequence independent" SNP?

Line 130: Table 1 can go to supplementary file unless readers are interested in individual cassava accessions.

Line 134: "Extraction of DNA and quantification of DNA... " can surely be shortened to "Extraction and quantification of DNA... "

Line 145 - 147: these are unnecessary details - who is interested to know which brand of 96-well PCR plate the samples were packed in?

Line 150 - 151: "As defined by Kilian et al. [38]" is just unnecessary wordiness. Just put the citation at the end of the sentence.

The whole DArT procedure paragraph section is text that needs to be replaced by citing the methods and not repeating them here. For example, the authors say that "genomic representation were generated following the procedure of [38]" and "A detailed procedure is documented by [37]" but still go into details of the laboratory protocols some of which are quite shallow and will not be helpful to the readers. Indeed the objective of the paper is to describe the diversity in the germplasm but not a paper on the genotyping methodology. So the latter should not be the highlight.

L161-162: What value does "Order:162 DCas18-3505 on 01/06/2018" add to the manuscript? How will this information benefit the reader?

L165 - 166: The way SNP calling is mentioned is not clear. "Markers were scored ‘1’ for presence, and ‘0’ for absence and

‘-’ for calls with non-zero count but too low counts to score confidently as “1”." Is this for SNPs or the presence absence DartTag?

L173 - 175: call rate - this level of details is unnecessary. Call rate is % another description of proportion of missing data which is self explanatory.

results:

L207 : which version of the cassava reference genome?

L210: How was reproducibility of the SNPs calculated and based on what?

L228 - 264: This whole section can be shortened by presenting the results graphically e.g. using histogram of pair-wise distance. This can be done for the silicone-dart and the regular SNPs.

L234 - 236: The critical distance threshold to declare whether two genotypes was based on only replicated DNA from 3 samples. In my opinion this is rather small sample size to make an inference about the threshold identifying clones. The authors need to produce a plot of the histogram of pair-wise distance and show if they can pick up the signal of the redundant DNA from these 3 samples.

L244: what is a "mutant" variety ?

The section needs to be made more concise.

L272 - 275: "This means that optimum number of groups that suits the distribution of similar cultivars inside the population was two, indicating that, two different groups (subpopulation1 and subpopulation2) contribute relevant genetic information across the population." should be "The optimum number of subpopulations is two". No need two be repetitive.

L278: How can the expected heterozygosity be close to zero in a non-inbred clonal crop like cassava?

Table 3: what does "inferred clusters" mean? The numbers seem to add up to 1 but is tis the proportion of ancestry to cluster 1 or 2???

L292: Why every time list all accessions in each cluster - or any given result section. This type of information should be in a table or figure for the interested reader to look into but serves no additional value to the manuscript.

Line 313: "The first two axes of the PCoA (Ofigure)..." It seems that the paper was not checked for submission because you cannot have a reference to a figure that is not numbered. The naming of figures/tables in the manuscript does not follow correct convention. Eg. "S3 Table" in Line 231 is actually not a supplementary table; "S1 and S2 Figs" in Line 299.

Discussion: Needs to be made more concise to increase readability.

Figures:

There are too many figures in the manuscript. Some can be joined together.

Figure 1: Represent the distribution of call rate using histograms and not pie-charts.

Figure Dendrograms. These aren't legible and need to be redrawn. There are tons of R-based tutorials on how to generate useful hierarchical clusters that are legible.

Figure 5: What is "average silhouettes" - ? These figures can be sent to supplementary files or be components of the STRUCTURE figure.

The PCA plots have wrong aspect ratio - even though the ranges of PC1 and PC2 are similar.

Figure 8. This plot can be provided as a supplimentary file. The mantel test are already provided in the results and this is redundant.

6. PLOS authors have the option to publish the peer review history of their article (what does this mean?). If published, this will include your full peer review and any attached files.

Reviewer #1: No

Reviewer #2: No

---

## [Author Response · Author response to Decision Letter 0]

27 May 2021

Response to editor and reviewers comments.

Comments Response

Editor’s comment: 

1. Please ensure that your manuscript meets PLOS ONE's style requirements, including those for file naming. The author has taken it into account. 

Please include a caption for figures 4-8 The authors have taken it into account. Caption for figures are indicated below the paragraph in which they were first mentioned

Reviewer #1 

Abstract 

Header: Take out additional symbols after number superscript These symbols are part of the title, author, affiliations formatting guidelines where asterisk* = corresponding author, Pilcrow (paragraph symbol) ¶ = first set of equal contributors, ampersand & = 2nd set of equal contributors etc.

Ln 1: Take out extra spaces throughout the document; just search for space space; Manihot esculenta The author has addressed this throughout the document

You need to say somewhere what the germplasm is? Landraces, breeding lines and from where? Before you get into results It has been addressed on line 29 and 30

Introduction 

Ln 52 due to its ability Corrected on line 50

Ln 53 Take out “than most staple crops” Corrected on line 51

Ln 54: contain Corrected on line 52

Ln56:] Corrected on line 54

Ln 58: Okogbenin and FrEgene – this work was a long time ago, and much has happened since then. Cassava is no longer an ‘orphan’ crop. Addressed on line 56

Ln 102: DArTSeq does not involve microarray. It purely involves complexity reduction (restriction digestion), and NGS. The authors need to be aware that before NGS came about, the company used a microarray based genotyping technique called DArT. This is not used much now, if at all. It seems the authors have got confused with the two techniques Addressed on line 97

Ln 106: I understand that SilicaDArT are a reflection of epigenetic status. They are epigenetic markers. The difference between DArTseq SNPs and Silico needs to be VERY clearly defined. Microarray is a technology, not a form of marker. It is not used in generating SilicoDArT Addressed on line 101

Ln107: Be consistent in how you define DArT-based SNP or DArTseq SNP Addressed on line 102 and throughout the text

Ln 109/110: With DarT there is no choice in restriction fragments. Xia et al 2004 in his work captured PstI/TaqI and PstI/ BstNI enzymes as rare and frequent cutters for cassava genome complexity reduction.

Ln 114: Be more specific about the genebank Addressed on lines 30 and 119

Materials and methods 

Ln120 eighty-seven should be 87 Addressed on line 117

Ln 121: How many were landraces and how many released cultivars? Addressed on line 118

Ln 124: twenty-five should be 25

 Addressed on line 121

Ln 127: This is the fist time you mention morphological. This should be mentioned in introduction.

Table 1: Why choose to report colour of apical leaves? Why dry matter? Also, cassava has storage roots, not tubers. (captured on ln . Addressed between line 109 and 110

Correction effected in the table 1

PCA among agro-morphological data identified apical leaves colour and DM as discriminating characters.

DArT procedure paragraph: Please be aware that DArT and DArTSeq are different procedures. There is no cloning in DArTSeq. This paragraph needs to be corrected Corrections effected between lines 145 and 149

Ln 164: You need to be clear that alleles were scored as 1 and 0; provided in two line format; or 0,1 and 2 as single row format with heterozygotes identified. This is critical. Distinguish scoring between Silico DArT and DArtSeq. Addressed between line 153 and 156

Ln 175: Don’t use DArT-based SNPs – these are DArTSeq SNPs Corrected throughout

Ln179: I would use the term duplicates rather than redundant. Addressed on line 168

Ln 178: You need to discuss why you used Gowers Gower’s like other distance measures can be used in germplasm classification by genebanks. Moreover, other similarity measures such as Dice and Euclidean were used for confirmation of duplicates

Ln 180: “pvclust” in R (include reference) Addressed on line 171 and between lines 674-675

Ln 182: SilicoDArT and DArTSeq SNPs Corrected on line 171

Ln 198: Why capital letters? I wouldn’t call it a peak; rather a flattening of the curve. Addressed on line 187, 189 and 264

Results 

Ln 207: What do you mean “18 haploid cassava genome”? Explain this. You mean chromosomes? 18 haploid chromosomes of cassava. Addressed between line 197 and 198

Fig 1a and b. These figures are not necessary.

Fig. 2 needs to be better quality

Fig 3. Axis labels should not be capitals. Bars must not overlap and needs better quality.

Fig. 4. Cannot read anything from this Figure.

None of the subsequent figures have numbers Call rate figure sent to supplementary

Fig 3 issues addressed in current fig 2.

Hierarchical clustering issues addressed in figure 3

Ln 253 : Use SilicoDArT throughout the document Correction effected through out

Ln 271: As far as I know you are looking for a flattening of the graph where deltaK changes, not a peak Addressed on line 264 and Fig 4

Ln 276: You need to define which accessions are in which subpop before you start talking about diversity. Individuals are allocated to sub-populations. Addressed between line 268 and 283

Ln 292: Give a reference to average linkage clustering. What is meant by this. Clarfy. Addressed on line 269 and 270

Ln 302: Three rather than 3; and I would call these putative landraces as they may actually be improved lines Addressed on line 279

Ln 304: all cassava is heterogeneous. This is not a good word. Perhaps use Admixture. Addressed on line 281

Ln 308: Is there a figure associatd with this?yes Addressed on line 284

Ln 316: I wouldn’t indicate left of right as the figure could be switched. Just give colour. Rather say cluster together… Addressed between line 309and 311

Ln 338: What do you mean 22,516 assigned to specific chromosomes? Addressed between line 331 and 332

Ln 341: I wouldn’t say these markers have to be developed. They are there, they just need to be visualized. Addressed on line 333

Ln 350: better genome coverage than what?? Than other classical DNA markers

Ln 353: What about Ferguson et al. (2019) and many studies by Rabbi et al. Emphasis was on application of classical DNA markers on some of these accessions from Ghana. Ferguson et al [68] and Rabbi et al [71] are considered throughout the discussion

Ln 369: This is not two DArT platforms – just two types of data are generated form one platform Addressed on line 362

Ln 408: two Addressed on line 407

Ln 411: Rather than heterogeneous, I would say admixture or relatively close relationship Addressed on line 410

Ln 416: The authors must understand the basic rules of reporting numbers; 1-10 should be written in full. Anything more than 10 is given in figures. Please correct throughout. Addressed through out.

It does not indicate anywhere whether the duplicates were taken out of the dataset before diversity analysis was conducted. This should be done This has been taken into between lines 298-302 and 410-449, however the authors wanted to use structure analysis to lay emphasis on those duplicates. 

Ln 429: You need to find out the background / pedigree of IITA material Most are advanced breeding lines.

Ln 444: You might want to say that DArTSeq SNPs are preferred as they are co-dominant which means heterozygotes can be determined. To reiterate consistency in results from both markers in diversity and conservational studies 

conclusions 

Ln 451: It is not an array that is used. Be consistent in naming the technologies throughout the doc.

 Addressed on line 459

Ln 459: I don’t think it is defined by geographical differentiation, but more by pedigrees – breeding lines vs landraces.

 Addressed on line 467

Reviewer #2 

1. While the study contributes to the body of knowledge about the diversity of the crop in Ghana, the authors repeatedly put forth statements that are rather outdated when justifying their study in the introduction section of the manuscript. For example, there is repeated mention that cassava is an orphan crop, with no genomic resources available and this is not correct (e.g. lines 28 - 30 and Lines 58 - 61). Indeed, one of the papers used to support this notion is almost two decades old (Okogbenin et al. Theoretical and Applied Genetics. 2003; 107:1452–1462.)

There are many high-quality reference genomes, hundreds of accessions have been whole-genome resequenced to produce hap-map, and several diversity analysis that used genotyping-by-sequencing have been published. Moreover, genomic resources have been used to map QTLs using biparental populations and GWAS. Moreover, genomic selection in cassava is quite advanced and comparable with other major crops. It is therefore not correct for the authors to say that there is very little progress in the crop. A rigorous literature review should address this misperception.

These are addressed on line 26 – 27, 56 -57

Line 42 - 45: seems like a concluding statement that should have been at the end of the abstract Correction effected between line 42 and 46

L58 - 61: fragmented sentence that is not understandable. What is the intended message? Addressed on lines 56 and 57

L80 - 81: "... as detailed by some authors" is not necessary. Similar types citation styles are in the manuscript. It should just be "The use of molecular tools in plant genetic analyses and crop improvement cannot be overemphasized [26-29]." Addressed on lines 77 and 78. Other corrections are effected through out.

L84 - 92: The authors introduce a description of other types of molecular markers, including anonymous markers such as RFLPs etc. What purpose does this serve? Back when SNPs were new, it was acceptable to describe its advantages compared with the traditional markers but this serves no purpose since SNP have become mainstream. Instead, the authors should describe DArTseq and compare it with other high-density SNP genotyping methods like GBS which has been extensively used for cassava. A comparison was been made between other marker systems that have been applied to some of these particular cassava germplasm and other works internationally. Four RAPDs were used by Asante and Offei 2003. SSR by Adjabeng 2020, on some of these cassava lines and others.

L96-97: Why would one want to use "sequence independent" SNP? The authors tried to shed a brief light on the classical DArT markers first applied in rice by Daccoud et al 2001.

Line 130: Table 1 can go to supplementary file unless readers are interested in individual cassava accessions. Been captured in the manuscript will enhance easy reference especially the structure analysis plot.

Line 134: "Extraction of DNA and quantification of DNA... " can surely be shortened to "Extraction and quantification of DNA... " Correction effected on line 131

Line 145 - 147: these are unnecessary details - who is interested to know which brand of 96-well PCR plate the samples were packed in? Addressed on line 141

Line 150 - 151: "As defined by Kilian et al. [38]" is just unnecessary wordiness. Just put the citation at the end of the sentence. The whole DArT procedure paragraph section is text that needs to be replaced by citing the methods and not repeating them here. For example, the authors say that "genomic representation were generated following the procedure of [38]" and "A detailed procedure is documented by [37]" but still go into details of the laboratory protocols some of which are quite shallow and will not be helpful to the readers. Indeed the objective of the paper is to describe the diversity in the germplasm but not a paper on the genotyping methodology. So the latter should not be the highlight Addressed between line 145 and 149

L161-162: What value does "Order:162 DCas18-3505 on 01/06/2018" add to the manuscript? How will this information benefit the reader? corrected on line 149

L165 - 166: The way SNP calling is mentioned is not clear. "Markers were scored ‘1’ for presence, and ‘0’ for absence and

‘-’ for calls with non-zero count but too low counts to score confidently as “1”." Is this for SNPs or the presence absence DartTag? Addressed between line 153 and 156

L173 - 175: call rate - this level of details is unnecessary. Call rate is % another description of proportion of missing data which is self explanatory. Correction effected on line 163

results: 

L207 : which version of the cassava reference genome? V61 and V8 model reference Cassava v6.1 and v8. Captured on line 198

L210: How was reproducibility of the SNPs calculated and based on what? The reproducibility data was available in the received data set from DArT P/L describing how reproducible the marking scoring was in replicated samples

L228 - 264: This whole section can be shortened by presenting the results graphically e.g. using histogram of pair-wise distance. This can be done for the silicone-dart and the regular SNPs Pair-wise distance between individuals are captured S3 table. Authors would like to maintain specific relationship among individuals. 

L234 - 236: The critical distance threshold to declare whether two genotypes was based on only replicated DNA from 3 samples. In my opinion this is rather small sample size to make an inference about the threshold identifying clones. The authors need to produce a plot of the histogram of pair-wise distance and show if they can pick up the signal of the redundant DNA from these 3 samples Only three samples were replicated in the shipped DNA for analysis.

Moreover, other dissimilarity measures such Dice for the bi-allelic silico and Euclidean were used in the verification of similarity measurement and identification of duplicates.

L244: what is a "mutant" variety ? it’s the only released mutant variety in Ghana developed following selection from artificially induced mutational lines.

L272 - 275: "This means that optimum number of groups that suits the distribution of similar cultivars inside the population was two, indicating that, two different groups (subpopulation1 and subpopulation2) contribute relevant genetic information across the population." should be "The optimum number of subpopulations is two". No need two be repetitive. Correction effected on line 265

L278: How can the expected heterozygosity be close to zero in a non-inbred clonal crop like cassava? Most of the duplicates were found in subpop1 which eventually led to a reduced expected heterozygosity of 0.08 but increased to 0.357 upon further analysis without the duplicates. 

Table 3: what does "inferred clusters" mean? The numbers seem to add up to 1 but is this the proportion of ancestry to cluster 1 or 2??? Overall proportion of membership of the sample in each of the two clusters.

Correction effected in table 3..line 293

L292: Why every time list all accessions in each cluster - or any given result section. This type of information should be in a table or figure for the interested reader to look into but serves no additional value to the manuscript Correction effected throughout however, few list are maintained for emphasis.

Line 313: "The first two axes of the PCoA (Ofigure)..." It seems that the paper was not checked for submission because you cannot have a reference to a figure that is not numbered. The naming of figures/tables in the manuscript does not follow correct convention. Eg. "S3 Table" in Line 231 is actually not a supplementary table; "S1 and S2 Figs" in Line 299. Corrections have been effected on line 307. 

S3 table is matrix of dissimilarity between individuals.

"S1 and S2 Figs are now supplementary figures for call rate distribution (S1 Fig) on line 197 (as suggested by reviewer #1) and bar plot of K=9 (S2 Fig) on line 277…

Discussions 

Figures:

There are too many figures in the manuscript. Some can be joined together Some joined and others sent to supplementary including information on call rate and mantel plot.

Figure 1: Represent the distribution of call rate using histograms and not pie-charts. Histogram generated for call rate, available in S1 Fig.

Figure Dendrograms. These aren't legible and need to be redrawn. There are tons of R-based tutorials on how to generate useful hierarchical clusters that are legible Changes in dendrogram presented in Fig 3.

Figure 5: What is "average silhouettes" - ? These figures can be sent to supplementary files or be components of the STRUCTURE figure. Correction effected

The PCA plots have wrong aspect ratio - even though the ranges of Aspect ratio (now 1:1) defect corrected in Fig 6a and 6b.

Figure 8. This plot can be provided as a supplimentary file. The mantel test are already provided in the results and this is redundant. Figure sent to supplementary in S3 fig.

---

## [Decision Letter · Decision Letter 1]

25 Jun 2021

PONE-D-21-06686R1

High-density DArT-based silicoDArT and SNP markers for genetic diversity and population structure studies in cassava (Manihot esculenta Crantz)

PLOS ONE

Dear Dr. Adu,

Thank you for submitting your manuscript to PLOS ONE. After careful consideration, we feel that it has merit but does not fully meet PLOS ONE’s publication criteria as it currently stands. Therefore, we invite you to submit a revised version of the manuscript that addresses the points raised during the review process.

We look forward to receiving your revised manuscript.

Kind regards,

Tzen-Yuh Chiang

Academic Editor

PLOS ONE

Journal Requirements:

Reviewers' comments:

Reviewer's Responses to Questions

**Comments to the Author**

1. If the authors have adequately addressed your comments raised in a previous round of review and you feel that this manuscript is now acceptable for publication, you may indicate that here to bypass the “Comments to the Author” section, enter your conflict of interest statement in the “Confidential to Editor” section, and submit your "Accept" recommendation.

Reviewer #1: (No Response)

2. Is the manuscript technically sound, and do the data support the conclusions?

Reviewer #1: Yes

3. Has the statistical analysis been performed appropriately and rigorously? 

Reviewer #1: Yes

4. Have the authors made all data underlying the findings in their manuscript fully available?

Reviewer #1: Yes

5. Is the manuscript presented in an intelligible fashion and written in standard English?

Reviewer #1: No

6. Review Comments to the Author

Reviewer #1: General: The manuscript still requires a major English revision

In. 33. Use expansion of PIC on first use, not on second use.

Ln 55. They are not tuberous roots, they are storage roots. Tubererous roots can sprout from eyes in the roots. Cassava roots do not do this, and are referred to as storage roots.

Ln 56 continue, not continuous

Ln 92. These are SNP markers visualized using DArTSeq technology. At least they should be called DArTSeq SNP markers, not just DArT markers.

Ln 105, what do you mean 1,000 candidate polymorphic clones. Are these distinct cassava clones. If they are, then why are they candidate? All cassava clones are polymorphic, so not necessary to state this.

159, 160: no need to repeat ‘polymorphic information content’ once it has been sued once. Also repeated on line 204.

Ln 292 How did you calculate Net nucleotide distance (add to methodology) and give a reference.

7. PLOS authors have the option to publish the peer review history of their article (what does this mean?). If published, this will include your full peer review and any attached files.

Reviewer #1: No

---

## [Author Response · Author response to Decision Letter 1]

8 Jul 2021

Response to editor and reviewer's comments.

Comments Response

Editor’s comment: 

1. Journal Requirements:

Please review your reference list to ensure that it is complete and correct. If you have cited papers that have been retracted, please include the rationale for doing so in the manuscript text, or remove these references and replace them with relevant current references. Any changes to the reference list should be mentioned in the rebuttal letter that accompanies your revised manuscript. If you need to cite a retracted article, indicate the article’s retracted status in the References list and also include a citation and full reference for the retraction notice The authors have taken them into account.

7. Okogbenin, Fragene was removed based on the previous suggestion by reviewer #1 (it was outdated).

9. Swaminathan MS was inserted following the review of literature

51. Maechler M, Rousseeuw P, Struyf A, Hubert M, Hornik K. (2013) was inserted as observantly pointed out by reviewer #1.

68. Ferguson ME, Shah T, Kulakow P, Ceballos H. A global overview of cassava genetic diversity. PLoS ONE. 2019 was inserted after reviewing the document as suggested by reviewer #1.

 The authors have taken them into account. 

The dimensions and resolution of the figures have been improved with Picture Analysis and Conversion Engine (PACE) to meet PLOS standards.

Reviewer #1 

General: The manuscript still requires a major English revision The authors have taken them into account. 

In. 33. Use expansion of PIC on first use, not on second use.

 It has been addressed on line 34.

Ln 55. They are not tuberous roots, they are storage roots. Tubererous roots can sprout from eyes in the roots. Cassava roots do not do this, and are referred to as storage roots.

 It has been addressed on line 55.

Ln 56 continue, not continuous It has been addressed on line 56.

Ln 92. These are SNP markers visualized using DArTSeq technology. At least they should be called DArTSeq SNP markers, not just DArT markers. The authors tried to shed a brief light on the classical DArT markers first applied in rice by Daccoud et al 2001. Referring to classical DArT markers)

Ln 105, what do you mean 1,000 candidate polymorphic clones. Are these distinct cassava clones. If they are, then why are they candidate? All cassava clones are polymorphic, so not necessary to state this. Corrected on line 105.

159, 160: no need to repeat ‘polymorphic information content’ once it has been sued once. Also repeated on line 204. Corrected throughout.

Ln 292 How did you calculate Net nucleotide distance (add to methodology) and give a reference It was an output from the Structure simulation analysis which has been referenced on line 179.

---

## [Editor Report · Decision Letter 2]

14 Jul 2021

High-density DArT-based silicoDArT and SNP markers for genetic diversity and population structure studies in cassava (Manihot esculenta Crantz)

PONE-D-21-06686R2

Dear Dr. Adu,

We’re pleased to inform you that your manuscript has been judged scientifically suitable for publication and will be formally accepted for publication once it meets all outstanding technical requirements.

Kind regards,

Tzen-Yuh Chiang

Academic Editor

PLOS ONE
---

## [Editor Report · Acceptance letter]

16 Jul 2021

PONE-D-21-06686R2 

High-density DArT-based SilicoDArT and SNP markers for genetic diversity and population structure studies in cassava (*Manihot esculenta* Crantz) 

Dear Dr. Adu:

I'm pleased to inform you that your manuscript has been deemed suitable for publication in PLOS ONE. Congratulations! Your manuscript is now with our production department. 

Kind regards, 

on behalf of

Dr. Tzen-Yuh Chiang 

Academic Editor

PLOS ONE